# Position: Contextual Integrity is Inadequately Applied to Language Models

Yan Shvartzshnaider [* 1]   Vasisht Duddu [* 2]

## Abstract

Machine learning community is discovering Contextual Integrity (CI) as a useful framework to assess the privacy implications of large language models (LLMs). This is an encouraging development. The CI theory emphasizes sharing information in accordance with *privacy norms* and can bridge the social, legal, political, and technical aspects essential for evaluating privacy in LLMs. However, this is also a good point to reflect on use of CI for LLMs. *This position paper argues that existing literature inadequately applies CI for LLMs without embracing the theory's fundamental tenets.*

Inadequate applications of CI could lead to incorrect conclusions and flawed privacy-preserving designs. We clarify the four fundamental tenets of CI theory, systematize prior work on whether they deviate from these tenets, and highlight overlooked issues in experimental hygiene for LLMs (e.g., prompt sensitivity, positional bias).

## 1. Introduction

The growing use of large language models (LLMs) to automate tasks across and within social contexts (e.g., workspace, households, health, education) raises questions about the models' privacy implications and effective ways to evaluate it. Traditional notions of privacy–focusing on restricting certain types of information, dichotomies of data types, data minimization, access control–provide inadequate protection in an inter-connected society (Nissenbaum, 2019). This has prompted efforts to incorporate a more robust notion of privacy that includes contextual and societal considerations, particularly for LLMs (Brown et al., 2022; Mireshghallah et al., 2023; Bagdasaryan et al., 2024; Li et al., 2024; Cheng et al., 2024; Fan et al., 2024).

Notably, these efforts evaluate privacy in LLMs using contextual integrity (CI), which defines privacy as the appropriate flow of information by adhering to *privacy norms*. CI provides a structured way to identify potential privacy violations based on the context (e.g., by capturing the actors' capacities in the information exchange, the information type, and the constraints of sharing information).

Despite the apparent simplicity of the CI framework, its application is far from trivial. A rote use of the CI framework, while insightful, does not advance our understanding of the privacy implications of LLMs. To meaningfully operationalize the CI theory, we require supporting its four essential tenets (Nissenbaum, 2019): **T1** Privacy is the appropriate flow of information; **T2** Appropriate flows should conform with privacy norms; **T3** Information flows are defined using five parameters; **T4** Ethical legitimacy of privacy norms is evaluated using the CI heuristic.

> **Position.** Existing LLM literature inadequately applies CI by not supporting the core CI tenets. This undermines the reliability of their claims, risking misinterpretations and flawed privacy-preserving system designs.

We would like to emphasize that the discrepancies in use of CI do not render the existing work obsolete. Nevertheless, the use of CI without adhering to the theory's main tenets sends the wrong message to the unfamiliar reader. Our goal is to set the record straight and to engage the ML community to help fulfill the full potential in applying the CI framework. We support our position by:

1. **Clarifying Tenets of CI Theory**. We specify the four fundamental tenets of the CI theory (**T1**-**T4**), and what constitutes the CI-based privacy analysis. (Section §2)

2. **Survey and Systematization of Prior Work**. We discuss nine "Alternate Views" of using CI for LLMs in prior works, and highlight how these works deviate from the CI core tenets. (Section §3)

3. **Experimental hygiene**. We highlight the importance of accounting for prompt sensitivity in LLMs (variation in responses due to small changes in prompts), and examine existing efforts. (Section §4)

*Equal contribution  [1]York University  [2]University of Waterloo. Correspondence to: Yan Shvartzshnaider <yansh@yorku.ca>, Vasisht Duddu <vasisht.duddu@uwaterloo.ca>.

*Proceedings of the 42nd International Conference on Machine Learning*, Vancouver, Canada. PMLR 267, 2025. Copyright 2025 by the author(s).

## 2. Contextual Integrity: Privacy in Context

Privacy is a loaded term that people generally to have a strong intuition about–if our privacy is violated, we feel some discomfort. However, this intuition has not been effectively translated into a definition that is useful to evaluate sociotechnical systems.

Over the years, scholars have debated different notions of privacy, and many definitions tend to be neutral or descriptive (Gavison, 1980), focusing on describing protected classes of information through some form of access control that defines the 'increase' or 'decrease' in the state of privacy, without considering the normative values of these various states. In practice, these notions of privacy are inadequate where information sharing is ubiquitous and on a large-scale (Nissenbaum, 2022).

In contrast to descriptive definitions of privacy, CI theory (Nissenbaum, 2009) embraces sharing information in appropriate and legitimate ways, in accordance with *privacy norms*. For privacy evaluation, CI relies on (a) the *descriptive framework* to detect norm-breaching information flows, and (b) the *CI heuristic* to perform a normative assessment of those breaches. The latter is *crucial* for evaluating disruptive technologies that, despite breaching existing norms, can still be appropriate if they align with social values, goals and contextual purposes.

Figure 1 summarizes the various steps for CI-based privacy analysis: Step ❶ (identifying the information flows), Step ❷ (identifying the established privacy norms), Step ❸ (conducting a breach analysis by checking for deviation from socially acceptable norms as a baseline), and Step ❹ (revisiting the legitimacy of a norm-breaching information flow by examining moral, political, and economic implications).

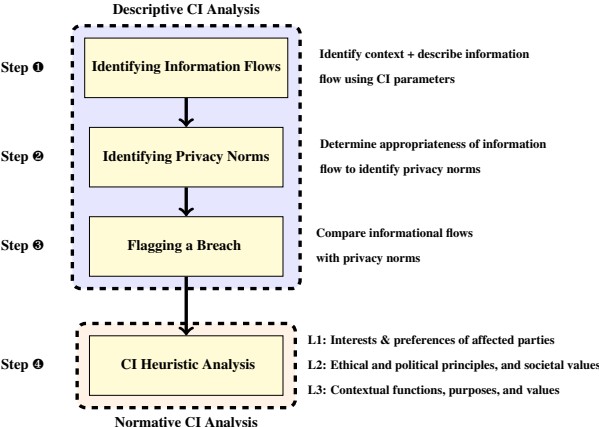

Figure 1: **Overview of CI Analysis.** Steps ❶-❸ is *descriptive analysis*; Step ❹ is *prescriptive or normative analysis*.

The CI theory is based on four fundamental tenets that en-

sure a rigorous privacy assessment with moral and ethical weight. These tenets convey the substantive assertions of CI while also allowing for comparisons with other definitions of privacy (Nissenbaum, 2019).

**T1 *Privacy* is the Appropriate *Flow* of Information**

This tenet establishes that privacy is not about secrecy (i.e., no information flow), but about ensuring the appropriate flow of information. According to CI theory, privacy violations result from inappropriate information flows, rather than from the protection of specific data types (e.g., those classified as personal, private, or sensitive). The concept of information flow is central to framing privacy in terms of the multiple dimensions of an information exchange, as defined by **T3** below. For instance, it is largely acceptable for a patient to share their sensitive health information with emergency services, confidentially (Nissenbaum, 2009).

**T2 Appropriate Flows Conform with *Privacy Norms***

This tenet defines the notion of appropriateness (left open by **T1**) by introducing "the construct of contextual informational norms that express or characterize information flows" Nissenbaum (2019).

The appropriateness of information flow is governed by established norms in different societal contexts. These norms "describe, prescribe, proscribe, and establish expectations for characteristic contextual behaviors and practices" (Nissenbaum, 2019). Social norms are in a constant state of flux. Many factors influence the norms that a local community or society at large adopts. The starting point of CI theory analysis is the perspective of established contextual norms. Although these norms can evolve and adapt in response to social, political, and cultural developments, CI analysis focuses on a snapshot of privacy norms in a given instance.

**T3 Define Information Flows using *Five Parameters***

This tenet defines the five essential parameters (also referred to as "CI parameters") that capture the information flow and norms: (i) roles or capacities of the actors (*senders*, *subjects*, and *recipients*) in the context they operate (like professors in an educational context and doctors in a health context); (ii) the *type of information* they share; (iii) *transmitted principle* to state the conditions under which the information flow is conducted. An example of information flow described using the five parameters is: *Patient* (sender) sharing *patient's* (subject) *medical data* (information type) with *a doctor* (recipient) *for a medical check up* (transmission principles). It is important that the values of the parameters are not arbitrary and they should reflect the contextual ontologies that define the roles, capacities, and information types of actors. *All five parameters are important, and failing to state any of them would*

*lead to an inconclusive outcome.*

The last point is crucial. It distinguishes works that refer to CI but do not align with privacy as defined by CI (as defined in **T1** and **T2**). Instead, these works use CI to re-frame other conceptions of privacy, such as dichotomies (e.g., public/private, personal/non-personal). In other words, they focus the analysis on individual parameters, like protecting specific information types or supporting privacy as control by prioritizing the transmission principle to enforce consent-based mechanisms.

### T4 CI Heuristic Assesses Ethical Legitimacy of Norms

This tenet provides the roadmap for a CI-based normative analysis. New information flows that challenge entrenched privacy norms—potentially leading to privacy violations–would require a closer examination through "a comparative assessment of entrenched flows against novel ones" in Step ❸ (Nissenbaum, 2015).

The CI heuristic (Step ❹) helps assess the ethical legitimacy of breached norms through several levels of analysis, considering ethical, political, societal, and contextual factors, and potentially consulting with relevant experts and professionals (Susser & Bonotti, 2024). The heuristic includes three levels:

- *Level 1* identifies the "winners" and "losers" in the new social reality, examining whose preferences and interests are affected, who gains, and what potential harms the new information flow may cause.
- *Level 2* examines societal values like justice and fairness, as well as political principles such as democracy and the rule of law, impacted by the new information flow.
- *Level 3* examines how the new information flows affect the context related values, functions and ends.

The CI heuristic relies on all the preceding tenets (**T1**-**T3**), without which multi-level heuristic analysis would not be possible. Deviating from any of the tenets results in the loss of a "fundamental insight of contextual integrity [that] information flows may systematically affect societal interests and values, making them important touchstones for normative evaluation" (Nissenbaum, 2015).

## 3. Alternative Views

We survey and systematize "Alternative Views" on the use CI in prior work. We contrast these prior works with our position, based on the CI tenets (**T1**-**T4**). We argue that existing works inadequately apply CI to LLMs, failing to embrace the theory's core principles, such as: (i) not subscribing to the CI's definition of privacy, instead relying on alternative formulations; (ii) incomplete definition of information flows by omitting any of the five CI parameters, essential for a comprehensive (unambiguous) analysis; (iii) failing to apply the CI heuristic to assess the ethical legitimacy of information flows, thereby neglecting CI's normative evaluation.

**Summary of Prior Work.** We identify nine works that have used CI theory to evaluate privacy in LLMs in two settings: *LLMs as Chatbots* (Gemini-Team, 2023; Reid et al., 2024; Dubey et al., 2024; Achiam et al., 2023), and *LLMs as Agents* to automate tasks like email composition, tool utilization, and form completion (Schick et al., 2024; Komeili et al., 2021; Gao et al., 2023; Parisi et al., 2022).

### *LLMs as Chatbots*

The proliferation of LLM-based chatbots in different contexts such as education, health, and software development has prompted the evaluation of the privacy implications of interactions between users and LLMs. LLM-based chatbots need to adhere privacy norms to ensure they appropriately share information in response to a query, specially when the norms vary drastically across cultures and societies. For instance, it might be appropriate to discuss human anatomy in an educational context, while the same discussion may be considered inappropriate in other contexts. We reviewed the following recent works that have used CI with LLM-based chatbots: (i) Mireshghallah et al. (2023) evaluate LLM responses to prompts with CI-based vignettes; (ii) "GoldCoin"(Fan et al., 2024) and "Privacy Checklist" (Li et al., 2024) approaches evaluate compliance of LLM-models with legal statutes like HIPAA; (iii) Ngong et al. (2024) rephrase users' prompts to prevent the sharing contextually inappropriate information in interactions between users and chatbots.

### *LLMs as Agents*

LLM-based agents raise additional ethical and moral questions about their role in our society. Unlike humans and human-led institutions, LLM governance is still in its early stage. As arbiters of information flows, it would be crucial for LLMs to adhere to societal norms and expectations.

The question of accountability and trustworthiness of these systems will play an key role in the adoption and deployment of these systems across different settings. Will society trust an LLM-led system to act on its behalf and align itself with contextual expectations? CI can provide valuable guidance in addressing these concerns.

The following works have used CI with LLM-based agents: Bagdasaryan et al. (2024) and Ghalebikesabi et al. (2024) developed an LLM agent that prevents the disclosure of private information. Bagdasaryan et al. (2024) defines private information as data irrelevant to the task and restricts the agent's access to only task-relevant information. Ghalebikesabi et al. (2024) designed information flow cards (IFC) to help LLM agents assess the appropriateness

of sharing information with an external third party. Similarly, Hartmann et al. (2024) consider the interaction between two models: a local model that needs to query a remote LLM for information about a specific task without including any personally identifying information that is not pertinent to the query context. Shao et al. (2024) propose a benchmark for evaluating LLM-based agents in various contexts, aiming to identify deviations in the model's responses and actions from established rules. Cheng et al. (2024) introduce another benchmark to evaluate agents' ability to correctly identify the relevant context and assess the appropriateness of information flow to avoid disclosing sensitive data in the model's responses.

**Systematization.** For each prior work, we mark alignment (✓) and deviation (✗) with a tenet in Table 1. We also summarize the main points of alignment in green and deviations in red . Our overview highlights the alternative view of CI, i.e., the interpretation of core CI tenets in prior works, and the discrepancies with the theory.

**T1: *Privacy* is the Appropriate *Flow* of Information**

Prior works use various interpretations of **T1**, often selectively chosen to support their specific objectives. They reframe the notion of privacy as leakage of "sensitive" and "private" data, as data minimization technique, or conflate "compliance violation" with "privacy violation." This often leads to a narrow interpretation of the theory to support specific tasks, focusing primarily on preventing leakage of certain data types evaluating the appropriateness of overall information flows. In other words, these works treat CI as a tool to enforce descriptive notion of privacy such as privacy as data minimization, public/private dichotomy, protection of sensitive data and procedural enforcement. These reframings go against the notion of privacy as contextual integrity that "does not accept the implications of other definitions that identify privacy with no flow, with stoppage, secrecy, and data minimization" (Nissenbaum, 2019).

**Public/Private Dichotomy.** Mireshghallah et al. (2023) do not fully adopt CI's privacy definition; use it instead to capture "private information leakage." Also, they incorporate other notions of privacy that focus on "discern[ing] private and public information." Similarly, Hartmann et al. (2024) use leakage of private entities (e.g., names and locations) as a proxy for evaluating appropriateness.

**Data Minimization.** Bagdasaryan et al. (2024) and Ngong et al. (2024) use an LLM to minimize the inappropriate sharing of information by the agent in the user-defined context. Bagdasaryan et al. (2024) focus on information flows that violates privacy directives such as "share information that can help with the task" or "share minimal information with third parties." Ngong et al. (2024) identify information flows containing "potentially sensitive information

in user prompts" between the users and LLMs. Similarly, Hartmann et al. (2024) refer to CI as a "data minimization privacy techniques" to prevent information flows with sensitive information.

**Compliance Violation.** Fan et al. (2024), Shao et al. (2024), and Li et al. (2024) view privacy violation as a breach of existing regulations, policies, and privacy case law. They conflate "compliance violation" with "privacy violation." Next section clarifies how legal statutes does not constitute privacy norms.

**Privacy Leakage.** Ghalebikesabi et al. (2024) and Bagdasaryan et al. (2024) evaluate "privacy leakage" that occurs when an LLM agent shares information with a third party that is not relevant to the task. Shao et al. (2024) also refer to "evaluation of privacy leakage in LM agents' actions." These conflate the descriptive and normative views of privacy, as discussed in Section 2.

According to CI, inappropriate information flow is not 'privacy leakage.' As discussed in **T2**, it would represent a *potential violation of established privacy norms*.

Hence, we mark Mireshghallah et al. (2023); Hartmann et al. (2024); Bagdasaryan et al. (2024); Ngong et al. (2024); Fan et al. (2024); Shao et al. (2024); Li et al. (2024); Ghalebikesabi et al. (2024) as ✗ for **T1**. Among the prior works, Cheng et al. (2024) align with **T1** in evaluating the ability of LLMs to assess the appropriateness of information flow with respect to established norms in a given context. Hence, we mark them as ✓ for **T1**.

> The deviation in **T1** lies in the different objectives outlined by prior work for using CI (measuring leakage of "sensitive" data, data minimization, or compliance violation), rather than assessing the appropriateness of information flow against privacy norms.

**T2: Appropriate Flows Conform with *Privacy Norms***

We identify two major deviations in **T2** in prior work in their definition of a *context* and *privacy norms*.

**Definition of Context.** Privacy, and especially ethical norms, carry moral weight in the overall functions, purposes, and ends of a social context. We note that many of the works refer to context in a very loose sense. CI views society as comprising multiple social contexts, each a distinct social sphere that includes (Nissenbaum, 2009): (i) *roles* that individuals or groups occupy within a context, such as a teacher, doctor, or student; (ii) *activities* relevant to the context and its functions (e.g., patient examination in the health context or delivering lectures in the education context); (iii) *norms* that govern acceptable actions and practices; (iv) *values* that guide actions within a context, such as promoting well-being and equity. In the

works we reviewed, context is primarily treated as a localized setting, focused on specific information flows for a given task, rather than as a broad social construct.

Bagdasaryan et al. (2024) consider a communication context that govern the LLM-agent's communication with the third-party services using user-defined "privacy directives." Cheng et al. (2024) arbitrarily define context by a domain (e.g., hospitality), intention (e.g., providing feedback), and interaction details (e.g., the user wants to plan a trip). Mireshghallah et al. (2023) define contexts as composition of "seed components" that describe a specific scenario information types, actors, and use. Fan et al. (2024) adopt the notion of context as defined by various legislations. They use contextual ontologies for the relevant roles but omit the contextual values and goals that privacy norms support, assuming that privacy legislation embodies these norms.

Ghalebikesabi et al. (2024) also follows this assumption in evaluating information flows based on how "a particular type of information might be regulated." Li et al. (2024) recognized the "discrepancies between the CI characteristics extracted from legal documents and those derived from real-world contexts." Nevertheless, they still focus on checking compliance with HIPAA.

Shao et al. (2024) although references privacy norms, fails to explicitly define context. They instead implicitly treating it as a set of rules outlined in privacy documents and crowdsourced through survey vignettes. Hartmann et al. (2024) define "local context" to describe "local models with access to sensitive data need to be equipped with a privacy-preserving mechanism that enables querying a remote model without sharing any sensitive data." Ngong et al. (2024) define context using a taxonomy of common user tasks in various domains such as health and wellness.

> Task specific definitions of 'context,' rather than a broad social construct, may lead to different—and often incorrect—interpretations of the notion of 'privacy norms.'

**Interpretation of Privacy Norms.** We observed a general ambiguity in how privacy norms are referenced in existing work. The focus of prior works is primarily on identifying explicit rules for enforcement, resembling rule-based systems, drawing inspiration from prior CI-based systems (Shvartzshnaider et al., 2019; Barth et al., 2006; DeYoung et al., 2010).

Bagdasaryan et al. (2024) and Ngong et al. (2024) highlight the complexity of defining privacy norms, describing it as an open issue in the literature. They opt for general "privacy directives," such as "share information that aids the task" or "share minimal information with third parties." Hartmann et al. (2024) also adopt a highly reductionist interpretation of privacy norms by defining the 'en-

tity leak metric,' which includes private data types, as a proxy for determining appropriateness. We now discuss prior work that use legal statues and crowdsourced preferences as proxy privacy norms.

*Legal Statutes are not Privacy Norms*

Laws, regulations, and policies can shape information flows that deviate from established cultural, societal, and moral norms. Like any new technology, these institutions can prescribe behaviors that challenge these norms. For example, while it may be legal to take a photo of someone on a train, doing so could potentially violate their privacy. Furthermore, cultural norms can vary across contexts and communities all of which may not be captured by legal statutes (Gerdon et al., 2020; Vitak & Zimmer, 2020; Utz et al., 2021). This is perhaps the clearest example of the difference between adhering to norms and ensuring legal compliance. Merely ensuring that a practice is legal does not necessarily make it privacy-respecting, according to CI. Communities may have different expectations for information handling practices that legal institutions are not sufficiently attuned to capture (Dworkin, 2013).

Fan et al. (2024), Shao et al. (2024), and Li et al. (2024) use legal statues, policy, regulation and privacy norms interchangeably. For instance, Shao et al. (2024) demonstrate the confusion in taxonomy by defining several "norm categories": legal norms (for information flows prescribed in policy and regulations) and unwritten norms (sourced from crowd sourcing, research papers). While privacy legislation and policies can prescribe and proscribe information flows, they might not reflect ethical and moral norms and therefore, could still violate privacy.

*Crowdsourced Preferences are not Privacy Norms*

There is a systematic conflation of privacy preferences and norms in many of the works. "Norms, especially ethical norms, may embody a great deal of wisdom; they may reveal majority expectations and settled accommodations among competing interests, but they may also reveal the oppressive victories of some interests over others" (Nissenbaum, 2022).

Contrary to other privacy definitions, CI views privacy as serving a social, collective good. Privacy norms may sometimes conflict with the interests of a few in favor of the greater good. For instance, the U.S. department of human health services requires physicians to report certain diseases. People may deem this as inappropriate as it compromises their confidentiality but it aligns with core public healthcare values (Nissenbaum, 2022).

In several works, privacy preferences were used as the ground truth to check for privacy violation. Building on existing CI methodologies, the efforts we examined

use user studies to gauge perceptions of appropriateness. Mireshghallah et al. (2023) relied on the Martin & Nissenbaum (2016)'s survey methodology to gauge appropriateness of information flows from a representative sample of community members.

While crowdsourcing methods can reveal overall perceptions of potential information flows through statistical tests and visualization techniques, it is important to emphasize that these survey methods should serve as a starting point to understand preferences, and further investigation into norms is required (see **T4**). Moreover, Martin & Nissenbaum (2016) purposely elicit individual preferences to compare with other non-CI-based studies.

Shao et al. (2024) presented participants with pre-filled vignettes containing values of some CI parameters—such as sender, recipient, and transmission principle—and asked them to fill in the missing data types and subject values of a contextual information norm. Ghalebikesabi et al. (2024) relied on eight annotators to rate the appropriateness of the information flow, explicitly aiming to gauge "expectations across the population (rather than individual preferences)." Cheng et al. (2024) use a subgroup of authors performed normative assessment of appropriateness of information flow scenarios. However, as the authors note, their annotations may not reflect norms.

Hence, we mark all prior works as ✗ for **T2** since they either use legal statutes or preferences as proxies for privacy norms to evaluate privacy violation.

> Existing works use proxies for privacy norms. While some acknowledge the discrepancy, they nonetheless treat these proxies as the ground truth.

### T3: Define Information Flows using *Five Parameters*

We examine whether prior work specify all five parameters whose values align with CI. The majority of the works we reviewed followed the five parameter template.

Bagdasaryan et al. (2024) tailors the CI parameters values to the LLM-agent tasks: *sender* is an LLM-agent that acts on behalf of the user, *recipient* is the third-party, *subject* is user's information, and *transmission principle* is stated in terms of "privacy directive" (describe constraints on information flow that "cover a range of user preferences," such as "share only relevant" or "share minimal information").

Shao et al. (2024) consider all five parameters where the *sender, recipient and transmission parameters* values ("privacy seeds") were from existing regulations HIPAA, FERPA, GLBA and *information types and subjects* were crowd-sourced to populate a CI-based vignette template from Shvartzshnaider et al. (2016).

Fan et al. (2024); Li et al. (2024) capture the values cor-

responding to legal policy and regulation ontologies, like HIPAA (Barth et al., 2006; DeYoung et al., 2010). Cheng et al. (2024) state all the five parameters where the values are based on synthetic dataset covering chat and email correspondences. Ngong et al. (2024) use parameter values from across different contexts such as health, finance, employment, politics, and religion among others.

However, Mireshghallah et al. (2023) did not use all five parameters and consider only *information type*, *actor*, and *use (transmission principle)*, using the template from Martin & Nissenbaum (2016): "Information about {information type} is collected by {actor} in order to {use}." But Martin & Nissenbaum (2016) use the three parameters while indicating that they deliberately simplified the task to improve legibility while acknowledging that a "full-blown operationalization of CI would have required five-factor vignettes based on the parameters ... critical to the definition of information privacy norms." Finally, Hartmann et al. (2024) do not mention the five parameters at all.

Hence, we mark Bagdasaryan et al. (2024); Li et al. (2024); Cheng et al. (2024); Shao et al. (2024); Fan et al. (2024); Ghalebikesabi et al. (2024); Ngong et al. (2024) as ✓ for **T3**, but Mireshghallah et al. (2023) as ✗.

> All prior works (except for Mireshghallah et al. (2023); Hartmann et al. (2024)) align with **T3** by identifying and using all the five CI parameters while describing information flows. Notably, they also state the correct values for the actors' CI parameters (i.e., sender, subject, recipient) representing roles and capacities.

### T4: CI Heuristic: Assess Ethical Legitimacy of Norms

The final tenet is crucial for the comprehensive CI analysis to evaluate privacy violations, and distinguishes this assessment from evaluating violation of rule compliance. Bagdasaryan et al. (2024), Ghalebikesabi et al. (2024), and Cheng et al. (2024) intentionally leave the discussion on the legitimacy of privacy norms outside the scope. Hartmann et al. (2024) do not consider the notion of appropriateness beyond reasoning about a set of private data types. Shao et al. (2024), Mireshghallah et al. (2023), Fan et al. (2024), and Li et al. (2024) view regulations and policies or crowdsourced preferences as ground truth for legitimacy. Hence, we mark all prior works as ✗ for **T4**.

> None of the works we evaluated even mention the CI heuristic, despite claiming to use CI, choosing instead to defer to laws, policies, and collective preferences as proxies for ethical, moral, or contextual legitimacy.

### Summary

Overall, we observed significant gaps between CI theory and its application for LLMs in current literature. We sum-

marize them in Table 1 covering the four CI tenets (**T1-T4**). Table 2 summarizes the alternative views, and contrast them to the CI tenets as our position. Partial adherence to the core principles of CI, or inadequate compliance, could lead significant privacy implications, and we highlight the negative implications in Table 2. We reflect on these results and recommend a path forward in Section 5.

Table 1: **Summary of Deviation from CI Tenets.** We indicate whether prior work aligns (✓) or deviates (✗) from the tenets (**T1-T4**).

| Literature | T1 | T2 | T3 | T4 |
|---|---|---|---|---|
| **LLM as a Chatbot** | | | | |
| **ConfAIde (Mireshghallah et al., 2023)** | ✗ | ✗ | ✗ | ✗ |
| **GoldCoin (Fan et al., 2024)** | ✗ | ✗ | ✓ | ✗ |
| **PrivacyChecklist (Li et al., 2024)** | ✗ | ✗ | ✓ | ✗ |
| **Hartmann et al. (2024)** | ✗ | ✗ | ✗ | ✗ |
| **Ngong et al. (2024)** | ✗ | ✗ | ✓ | ✗ |
| **LLM as an Agent** | | | | |
| **Airgap (Bagdasaryan et al., 2024)** | ✗ | ✗ | ✓ | ✗ |
| **PrivacyLens (Shao et al., 2024)** | ✗ | ✗ | ✓ | ✗ |
| **Ghalebikesabi et al. (2024)** | ✗ | ✗ | ✓ | ✗ |
| **CI-Bench (Cheng et al., 2024)** | ✓ | ✗ | ✓ | ✗ |

## 4. Experimental Hygiene for LLMs

In addition to adherence to core CI tenets, we also examine the validity of the experiments for using CI-based methodologies in LLMs. A recent study (Shvartzshnaider & Duddu, 2025) shows that responses to CI-based prompts on information flow appropriateness vary with paraphrasing or changing Likert scale positions. This emphasizes the need to consider non-adversarial robustness in LLM responses for reliable conclusions.

We highlight the following sources of variation in LLM responses: (i) same prompt sensitivity, (ii) paraphrased prompts sensitivity, (iii) 'position bias' sensitivity.

### Same Prompt Sensitivity

LLMs give different responses to the same question, especially when varying the temperature parameter, as it controls the extent of creativity in the responses. Therefore, it is important to properly account for variation in responses when the same prompt is queried multiple times.

Several works control for variation in responses for the same prompt, by averaging multiple responses (e.g., Mireshghallah et al. (2023); Ghalebikesabi et al. (2024); Shao et al. (2024). Alternatively, setting the temperature parameter to zero results in deterministic responses.

### Paraphrasing Prompt Sensitivity

LLMs are notoriously sensitive to small changes in prompts. Previous studies on LLMs have shown that benchmark results are unreliable if they do not account for

response variation (Cao et al., 2024; Errica et al., 2024; Lu et al., 2024; Gan & Mori, 2023; Cao et al., 2024; Sclar et al., 2024; Loya et al., 2023; Hida et al., 2024), leading to potentially incorrect conclusions in evaluating LLM-based systems (including for CI-based analysis).

> None of the prior works (Mireshghallah et al., 2023; Ghalebikesabi et al., 2024; Shao et al., 2024) have accounted for prompt variation in LLM responses.

### Position Bias Sensitivity

Prior works have shown that LLMs tend to exhibit a bias towards specific options in multiple-choice questions (Zheng et al., 2023; Zhang et al., 2024; Hsieh et al., 2024; Yu et al., 2024; Shi et al., 2024). For example, a model may consistently prefer the first option, even after changing the order of the options. Position bias suggests that the LLMs are not responding based on "reasoning," but rather selecting the most likely response, which is biased towards a specific option. Since all prior works prompt the LLMs to identify the appropriateness of information flows, position bias towards a specific option could result in incorrect evaluation and conclusions–such as allowing an inappropriate information flow due to bias toward a specific position indicating appropriateness.

> None of the prior works account for variation due to position bias which could lead to incorrect conclusions.

**Summary** None of the prior works discuss all aspects of non-adversarial robustness that could lead to unreliable and incorrect conclusions. Existing techniques to help account for prompt sensitivity (Mizrahi et al., 2024) and position bias (Zheng et al., 2023; Zhang et al., 2024; Hsieh et al., 2024; Yu et al., 2024; Shi et al., 2024), can help make the evaluation more robust. For example, to minimize "same prompt sensitivity," we can use the K-shot prompting technique (Zhuo et al., 2024). Another possible approach is to average the responses from multiple prompt variants to reduce the influence of outliers (Mizrahi et al., 2024). Additionally, we can employ a multi-prompt assessment strategy that takes the majority or super-majority of responses across multiple prompt variants (Shvartzshnaider & Duddu, 2025).

## 5. Discussion

CI framework is intuitive but not easy-to-implement. This often leads to its use without fully considering the key principles involved in verifying the legitimacy of norm-breaching information flows with respect to contextual functions, values, and goals. These are important because, according to CI, *privacy has a social value*. Privacy does not depend on individual preferences, procedural oversight,

Table 2: **Summary of alternate views from prior works, their negative implications, and our position.**

| CI Tenet | Alternate Views | Implication | Our Position |
|---|---|---|---|
| **T1** | Using CI to enforce: (i) privacy as data minimization, (ii) private/public dichotomy, (iii) protection of specific categories such as personal, sensitive, task-relevant data. | Privacy violations under these definitions may not be considered violations within the CI framework. | Privacy is defined and violated on the basis of appropriateness of the flow of information, rather than the type of data, procedural compliance or the extent of sharing. |
| **T2** | Using proxies for privacy norms to evaluate privacy violations: (i) privacy policies and literature, (ii) legal statutes, (iii) crowdsourced preferences. | Relying on inadequate proxies for privacy norms could lead to incorrect assessments of the appropriateness of information flows. | Governing institutions can be designed to support and reflect privacy norms that align with contextual values and goals, but this is not always the case. |
| **T3** | Describing the CI flow without 
 • indicating the values for all five CI parameters 
 • using roles and capacities to state the values for the actor parameter | Not specifying values of CI parameters introduces ambiguity in assessing privacy implications. | Omitting or providing a vague parameter description results in an inconclusive CI analysis, as the parameters reflect the underlying structure and relationships within the flow's context. |
| **T4** | Designing sociotechnical systems to arbitrate or generate novel information flows without addressing how to assess the legitimacy of the practice. | Without CI heuristic, information flows for novel technologies, initially be flagged as inappropriate, could be incorrectly identified as breaches, despite their potential societal benefits. | CI heuristic provides key guidelines for assessing norm-breaching flows that may contribute to societal values and goals. |

and ensuring the "sensitive" or "private" data is protected.

**Subscribing to the CI theory.** Applying CI envisions a world where the governing institutions–such as law, policies, regulation, ethical codes and practices–ensure appropriate and legitimate flow of information in accordance with the privacy norms. It involves explicitly considering the role of the system within a broader context and recognizing the sociotechnical implications of proposed solutions. As a social theory, applying CI requires more than a purely algorithmic approach. In the context of LLMs, we are required to incorporate societal processes such as governance, policy, and legal institutions, cultural aspects, to determine appropriateness of information flows.

Achieving consensus on privacy norms (e.g., for **T2**) might require a deliberate process that involves discussions among various expert groups, such as academic researchers, government representatives, and industry professionals (Susser & Bonotti, 2024). We would require a similar deliberative process where any novel or norm-breaching flows need to be examined based on their merits and contribution to the social values, contextual functions and goals, with the help of the CI heuristic for **T4** (Level 1, 2, and 3 analysis in (Section 2)).

**Source of confusion.** Inadequate applications of CI are not exclusive to LLMs. Part of the issue stems from a lack of consistency in the multidisciplinary literature and in picking the right authoritative source[1]. As Benthall et al. (2017)

observed, the multidisciplinary nature of CI applications leads to different interpretations or conflations of the theory's tenets, such as the meaning of social context and privacy norms. These are often overlooked outside the social sciences. This is not necessarily a bad thing. Like all theories, the implementation may rely on real-world assumptions that may not be as comprehensive as the theory requires–such as treating the LLM-agent as a sender without specifying its capacity or role within a given context. Furthermore, some parts of the framework may be useful outside the scope of the overarching theory, such as using the CI framework to check for compliance or gauge user preferences. In these cases, the authors should exercise caution in their claims: they do not preserve privacy as defined by CI but take inspiration from CI to preserve privacy according to other descriptive theories (e.g., data minimization or personal data protection).

**Conclusion.** While this may seem like a matter of semantics to some, as sociotechnical systems are integrated into our society, governance mechanisms will depend on this distinction. Failing to uphold, or clearly stipulate, the core CI tenets makes claims of using CI to evaluate privacy, at best, inaccurate, and at worst, misleading and potentially harmful. We call on the LLM community to work closely with information and social scientists to better understand the roles LLMs should play in shaping and governing our information.

# Impact Statement

This position paper presents work aimed at advancing the field of machine learning by providing guidance on adopting CI as a social and meaningful conception of privacy. The paper challenges current efforts in operationalizing the theory, highlighting discrepancies that arise when its core tenets are disregarded. We emphasize that while the mis-

---

[1]Over the years, the theory has evolved since its introduction in the Washington Post article (Nissenbaum, 2004). The most accurate account is the *Privacy in Context* book (Nissenbaum, 2009). Some works cite the Washington Post article, but not the book (Shao et al., 2024; Fan et al., 2024; Hartmann et al., 2024). Ghalebikesabi et al. (2024) cite the book, while Bagdasaryan et al. (2024), Cheng et al. (2024), Mireshghallah et al. (2023), and Ngong et al. (2024) cite both.

aligned use of the CI framework may offer some benefits, the theory itself conveys a deeper message than the simplistic, rule-based pattern-matching tasks it is often reduced to. CI views privacy as a social and public good. Its core tenets assert that modern society should embrace information sharing in accordance with established privacy norms, providing a robust framework for evaluating the ethical and moral weight of using LLM-based systems in our society.

## Acknowledgments

We acknowledge the support of the Natural Sciences and Engineering Research Council of Canada (NSERC), RGPIN-2022-04595. Vasisht is supported by IBM PhD Fellowship, David R. Cheriton Scholarship, and Cybersecurity and Privacy Excellence Graduate Scholarship.

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
