# OpenReview forum: "Position: Contextual Integrity is Inadequately Applied to Language Models"
_ICML.cc/2025/Position_Paper_Track — ICML 2025 Position Paper Track poster_

### Official Review · Reviewer_yTd7 · 2025-03-12

**Significance:** 3
**Argument Clarity:** 3
**Rating:** 4
**Confidence:** 1

**Questions:**

N/A. I am not familiar with contextual integrity or the related literature. I strongly recommend that the AC seek opinions from other reviewers.

**Discussion Potential:**

3

**Paper Summary:**

The authors suggest the existing literature often lacks support for core contextual integrity (CI) tenets, leading to “CI-washing. The authors aim to clarify misconceptions and  promote the CI framework’s rigorous application within the ML community.

**Position:**

Yes

**Position In Title:**

No

**Related Work:**

3

**Strengths And Weaknesses:**

The position is supported with sufficient evidence.

**Support:**

3

---

> ### Author Rebuttal · Authors · 2025-03-28
>
> We appreciate your “accept” recommendation. We try to provide sufficient background on contextual integrity (Section 2) to help non-experts understand the concepts, and a systematization of existing literature discussing a deviation from the theory. Additionally, we have proposed to update parts of the paper in response to some suggestions from other reviewers (Reviewer NbHH, UM5x, DYeR) that could be helpful. Is there anything we can clarify for you?

---

### Official Review · Reviewer_UM5x · 2025-03-15

**Significance:** 3
**Argument Clarity:** 3
**Rating:** 4
**Confidence:** 4

**Questions:**

See above

**Discussion Potential:**

4

**Paper Summary:**

Contextual Integrity (CI) is gaining significant attention as a privacy framework for evaluating and protecting user privacy in large language models (LLMs). However, recent works applying CI often fail to fully incorporate all necessary aspects of the framework. This paper examines existing literature to identify the specific components of CI that are frequently overlooked.

**Position:**

Yes

**Position In Title:**

Yes

**Related Work:**

3

**Strengths And Weaknesses:**

Overall, this work is valuable and has the potential to have discussion about the future of privacy in language models.. In my opinion it will also help to develop a more robust privacy evaluation framework for language models. Given the increasing number of paper in this field, this paper can be a  valuable guidance for better evaluations in future studies.

To improve the impact, the paper could benefit from including concrete examples of potential negative consequences arising from the incomplete application of CI principles, particularly within the context of existing studies.

General concern is the potential difficulty in adhering to all aspects of CI. Given its reliance on norms and heuristics in general it would be hard to make sure all aspects of the CI is fully studied but the existing literature do not even consider some of them.

While using terminology like CI-washing can be very effective to spark discussion but it can be perceived as overly critical.

**Support:**

3

---

> ### Author Rebuttal · Authors · 2025-03-28
>
> We thank you for your insightful (and encouraging) review. We would like to clarify some points raised in your review and indicate our proposed actions. Below, your comments are in bold.
>
> **Negative consequences arising from the incomplete application of CI** (similar response to Reviewer DYeR)
>
> *Response*:  We identify the following negative implications of not following each of the CI tenets:
> - T1: “Privacy as contextual integrity does not accept the implications of other definitions that identify privacy with no flow, with stoppage, secrecy, and data minimization” [1]. As an implication, a violation according to other definitions of privacy may not be considered a violation by CI.
> - T2: Using inadequate proxies for privacy norms, could lead to incorrect determination of appropriateness of information flows.
> - T3: Not fully specifying the values of the parameters introduces ambiguity in determining their appropriateness (e.g., as shown in [2]).
> - T4: Without CI heuristic, information flows corresponding to novel technologies, which may be appropriate and beneficial to the society, would be flagged as norm-breaching.
>
> *Proposed Changes*: We will add the above description under a new sub-heading “Implications” in Section 5.
>
>
> **Difficulty in adhering to all aspects of CI** (similar response to Reviewer NbHH)
>
> *Response*: Yes, we agree with your characterization that it is a "general concern is the potential difficulty in adhering to all aspects of CI" specially when "the existing literature do not even consider some of them". We plan to include some recommendations to implement various tenets.
>
> CI is a social theory. Hence, we speculate that applying CI to LLMs would require more than a purely algorithmic solution. We require incorporating societal processes such as  governance, policy, and legal institutions, and cultural inclusivity, to determine appropriateness of information flows. “Properly” implementing CI would require a multidisciplinary effort (as mentioned in Section 5). This entails explicitly thinking about the role the system plays in the greater context to protect and respect privacy, sociotechnical implication of their solutions.
>
> We will also highlight recommendations to meet T2 and T4 which are challenging:
> - T2: For gauging  privacy norms, we suggest a deliberative process described in [3] inspired by political science, which involves different expert groups like academic experts, government representatives, and industry professionals in  conversations to reach consensus on social privacy norms. The goal is to assess existing privacy norms, evaluate the impact of new data technologies on these norms, and determine necessary interventions or regulations.
> - T4: Once a breach is flagged in Step 3 (Figure 1), we can use CI Heuristic by deliberation among relevant domain experts on the merits and contribution of the breaching information flow by considering level 1, 2, and 3 analysis as described under T4 in Section 2.
>
> *Proposed Changes*: We will include the above discussion as a new subheading in Section 5.
>
> **“CI-Washing” is overly critical** (similar response to Reviewer NbHH)
>
> *Proposed Changes*: We can soften the title to “Contextual Integrity is Inadequately Applied to Language Models”. Will that help resolve your suggestion?
>
> **[Citations:]**
>
> [1] “Nissenbaum, Helen. "Contextual integrity up and down the data food chain." Theoretical inquiries in law 20.1 (2019): 221-256.”
>
> [2] Martin, Kirsten. "Privacy notices as tabula rasa: An empirical investigation into how complying with a privacy notice is related to meeting privacy expectations online." Journal of Public Policy & Marketing 34.2 (2015): 210-227.
>
> [3] Susser, Daniel, and Matteo Bonotti. "Privacy Mini-Publics: A Deliberative Democratic Approach to Understanding Informational Norms." Annual Symposium on Applications of Contextual Integrity. 2024. Link: https://privaci.info/symposium5/papers/Privacy-Mini-Publics-CI%20Symposium-Daniel%20Susser.pdf)

---

### Official Review · Reviewer_DYeR · 2025-03-15

**Significance:** 3
**Argument Clarity:** 4
**Rating:** 3
**Confidence:** 4

**Questions:**

Please kindly refer to the Weaknesses.

**Discussion Potential:**

3

**Paper Summary:**

Large language models (LLMs) are increasingly used in various social contexts, raising concerns about their privacy implications. The theory of Contextual Integrity (CI) is being adopted to evaluate privacy in LLMs, as traditional privacy conceptions are inadequate. CI defines privacy as the appropriate flow of information based on privacy norms and offers a structured way to identify potential privacy violations. However, the paper argues that existing literature often engages in "CI-washing," using the CI framework for LLMs without fully adhering to its four fundamental tenets: T1 (privacy is the appropriate flow of information), T2 (appropriate flows conform to privacy norms), T3 (information flows are defined by five parameters), and T4 (ethical legitimacy of privacy norms is evaluated by the CI heuristic). CI-washing can lead to incorrect conclusions and flawed privacy-preserving designs. To support its position, the paper clarifies the tenets of CI theory, surveys and systematizes prior work to show how it deviates from the core tenets, and highlights the importance of accounting for prompt sensitivity in LLMs as part of experimental hygiene. The paper concludes by emphasizing that CI provides a rigorous privacy evaluation framework, that "CI-washing" is not unique to LLMs, and calls for the LLM community to collaborate with information and social scientists. It also stresses the need to investigate non-adversarial robustness in CI-based approaches for evaluating LLMs.

**Position:**

Yes

**Position In Title:**

Yes

**Related Work:**

4

**Strengths And Weaknesses:**

Strengths:
* The paper effectively identifies the problem of "CI-washing" in the application of CI theory to LLMs. By pointing out that existing literature often fails to fully embrace the core tenets of CI, it highlights a significant gap in the current research on privacy evaluation for LLMs.
* The paper conducts a systematic analysis of the CI theory and its application to LLMs. It clearly outlines the four fundamental tenets of CI and then surveys and systematizes prior work to show how it deviates from these tenets.
* The call for the LLM community to work closely with information and social scientists is a strong point. Given the complex nature of privacy in the context of LLMs, which involves social, legal, and ethical aspects, interdisciplinary collaboration is essential. This recommendation can potentially lead to more comprehensive and accurate privacy evaluations for LLMs.

Weaknesses:
* The term CI-washing is not formally defined. As the main position target of this paper, the concept of CI-washing is very vague as it only appears in the Introduction and the Discussion and Conclusion sections. This vagueness makes it difficult to grasp the main position of this paper.
* This paper could benefit from more practical examples. Concrete examples of how "CI-washing" has led to incorrect privacy conclusions or flawed designs in real-world LLM applications would make the argument more tangible and easier to understand for readers who may not be familiar with the CI-related concepts.
* Although the paper highlights the importance of accounting for prompt sensitivity in LLMs as part of experimental hygiene, it does not provide specific solutions or suggestions on how to address this issue. Empirical evaluation of prompt sensitivity could also lead to more concrete evidence supporting this importance.
* The paper mentions that CI-based approaches for evaluating LLMs lack thorough testing in probabilistic language models and that the models' responses can vary based on configurations, but it does not elaborate on how to ensure non-adversarial robustness or what specific steps should be taken to address these issues.

**Support:**

2

---

> ### Author Rebuttal · Authors · 2025-03-28
>
> We thank you for your insightful (and encouraging) review. We would like to clarify some points raised in your review and indicate our proposed actions. Below, your comments are in bold.
>
> **“The term CI-washing is not formally defined”**
>
> *Response*:  We refer to “CI washing” to works that claim to use CI but do not subscribe to it four necessary tenets. This includes:
>
> - *Reformulating existing privacy definitions* without aligning with the definition of privacy according to CI's core principles.​
> - *Incomplete definition of information flows* by omitting any of the five CI parameters—sender, receiver, subject, information type, and transmission principles—which are essential for a comprehensive (unambiguous) analysis.​
> - *Failing to apply the CI heuristic* to assess the ethical legitimacy of information flows, thereby neglecting the framework's normative evaluation process.
>
> *Proposed Changes*: We will highlight the above description of CI washing more prominently at the beginning of Section 3. Will adding the above description help resolve your concern?
>
> **Note:**
>
> In response to the feedback from other reviewers, we are considering "Inadequate Application of CI to LLMs" instead of "CI-Washing" to reflect that CI is not fully operationalized for LLMs. Regardless, we will add a formal description of the "inadequate application" as suggested above.
>
> **Concrete examples of the implications of CI-Washing**
>
> *Response*:  We identify the following negative implications of not following each of the CI tenets:
> - T1: “Privacy as contextual integrity does not accept the implications of other definitions that identify privacy with no flow, with stoppage, secrecy, and data minimization” [1]. As an implication, a violation according to other definitions of privacy may not be considered a violation by CI.
> - T2: Using inadequate proxies for privacy norms could lead to incorrect determination of appropriateness of information flows.
> - T3: Not fully specifying the values of the parameters introduces ambiguity in determining their appropriateness (e.g., as shown in [2]).
> - T4: Without CI heuristic, information flows corresponding to novel technologies, which may be appropriate and beneficial to the society, would be flagged as norm-breaching.
>
> *Proposed Changes*: We will clarify the above implications under a new sub-heading in Section 5.
>
>
>
> **Empirically demonstrating non-adversarial robustness (e.g., prompt sensitivity)**
>
> *Response*: A recent work empirically demonstrates how the responses for CI-based prompts about the appropriateness of information flows, vary with paraphrasing or changing the position of likert scale options [3].
>
> *Proposed Changes*: We will add the above clarification at the beginning of Section 4 and include citation [3].
>
>
>
> **Recommendations for non-adversarial robustness (e.g., prompt sensitivity)**
>
> *Response*: We identify the following recommendations to minimize prompt sensitivity:
> - Take the average over responses from multiple prompt variants (paraphrasing + changing the Likert Scale) to remove the influence of outliers [4]. This is useful for minimizing “same prompt sensitivity”.
> - We can use a multi-prompt assessment strategy where we take the majority or supermajority over the responses corresponding to different prompt variants [3].
> - K-shot prompting strategies (i.e., including example prompts and corresponding responses) has been shown to reduce prompt sensitivity [5].
>
> *Proposed Changes*: We will add the above description under a heading “Recommendations” in Section 4. Will that resolve your suggestion?
>
>
>
> **[Citations]**:
>
> [1] Nissenbaum, Helen. "Contextual integrity up and down the data food chain." Theoretical inquiries in law 20.1 (2019): 221-256.
>
> [2] Martin, Kirsten. "Privacy notices as tabula rasa: An empirical investigation into how complying with a privacy notice is related to meeting privacy expectations online." Journal of Public Policy & Marketing 34.2 (2015): 210-227.
>
> [3] Shvartzshnaider and Duddu. Investigating Privacy Bias in Language Models. arXiv. 2025.
>
> [4] Mirzahi et al. State of what Art? A call for Multi-Prompt LLM Evaluation. TACL 2024.
>
> [5] Zhuo et al. ProSA: Assessing and Understanding the Prompt Sensitivity of LLMs. EMNLP 2024.

---

> > ### Comment · Reviewer_DYeR · 2025-04-03
> >
> > Thank you for the response and I have increased my rating accordingly.

---

### Official Review · Reviewer_NbHH · 2025-03-25

**Significance:** 3
**Argument Clarity:** 3
**Rating:** 4
**Confidence:** 4

**Questions:**

1. The authors have adopted "CI washing" which is a strong term IMO probably because they want to keep the title provocative. But CI is also a framework that cannot be easily implemented in its full glory in practice. It is a road towards progression and for this reason it may be good for authors to consider gradations of implementations of CI tenets rather than a binary "Yes" or "No".

In a related vein, are notions of privacy such as “data minimization” subsumed by CI? If so, the position of the paper could be more nuanced. May be it can argue that the current works claiming to have CI are using a more restrictive notion of it.

2. The authors could modify the title that clearly states their position that current literature does not implement CI fully ("CI washing for language models" does not imply this directly).

3. T1 and T2 seem quite close to each other. Can the authors use an example to clearly delineate this during the definition of them?

4. Can the authors suggest some concrete recommendations for overcoming some of the hurdles in implementing the tenets? Or may be they can also discuss if it is possible at all to implement some of them in the real world with reasonable effort. Should this system then involve human-in-the-loop and not be purely algorithmic, as in the literature we have now?

5. In the conclusion, the authors mention that the CI framework is intuitive and easy-to-implement. It is certainly intuitive, but is it easy to implement? The review of the authors suggest that it is not.

**Discussion Potential:**

4

**Paper Summary:**

The authors of the paper argue that current literature that has adopted the privacy notion of contextual integrity (CI) have not embraced the theory's fundamental tenets. The authors discuss the four tenets of the theory (T1-T4) and show how the existing works miss most of these tenets. The authors claim that this amounts to a form of CI-washing. They also highlight issues of experimental hygiene which may also undermine the claims of the current works. The authors conclude by discussing the potential harms of CI washing and a call to action.

## update after rebuttal
I am convinced with the author responses and am updating my rating accordingly in the hope that they will implement the proposed changes.

**Position:**

Yes

**Position In Title:**

No

**Related Work:**

3

**Strengths And Weaknesses:**

The authors have proposed four key tenets of CI theory that they use to systematize the prior work. While I am not familiar with all the prior work, the claims made by the authors on the specific tenets seem mostly reasonable. Another strength of this work is analysis of both "LLMs as Chatbots" and "LLMs as Agents". The tenets are generally clear although I have some questions (see questions below). The alternative views are generally clear and their connection to the CI tenets is made well. The discussion on experimental hygiene is motivated well. In addition, the call to action provided by the authors in the concluding section where they discuss the importance of having good mechanisms for sociotechnical systems is appropriate.

Some of the things I would need more clarity on include the arguments around CI washing, the rubric in Table 1, and perhaps inclusion of more concrete recommendations for implementation of the tenets. See also my questions.

**Support:**

3

---

> ### Author Rebuttal · Authors · 2025-03-28
>
> We thank you for your insightful (and encouraging) review. We would like to clarify some points raised in your review and indicate our proposed actions. Below, your comments are in bold.
>
> **“CI-Washing” is a strong term**
>
> *Response*: We picked the term to indicate that prior works do not subscribe with CI core tenets and hence do not actually rely on CI's key principles in their design. The title is not a clickbait, and we meant to foster constructive discussions on this topic, as experts in CI. It is not our intention to diminish the merits of prior work.
>
> *Proposed Changes*: We can soften the title to “Contextual Integrity is Inadequately Applied to Language Models”. Will that help address your concern?
>
> **Title does not state a position**
>
> *Proposed Changes*: As indicated above, changing the title “Contextual Integrity is Inadequately Applied to Language Models” clearly states our position.
>
> **Credit authors for gradations of implementation of CI tenets?**
>
> *Response*: As we state in the paper, it is not our intention to dismiss existing, valuable work. CI is a well-defined theory which requires a complete implementation of the four core tenets.  This is not optional. Hence, we push back on claims in existing works that their systems subscribe to privacy according to the CI theory. These works could explicitly indicate the aspects *inspired by CI* while clarifying their *deviations* from the fundamental theory (as mentioned in Section 5).
>
> *Proposed Changes*: This is important to clarify in Section 5. Will that resolve your suggestion?
>
> **Is data minimization subsumed by CI?**
>
> *Response*: This is a core issue of one of our arguments. Privacy as CI, explicitly states that it is not about minimizing data sharing, rather focus on appropriateness of information flows. Hence, “data minimization” is not subsumed by CI and it is a different notion of privacy which is not covered as part of CI. “Privacy as contextual integrity does not accept the implications of other definitions that identify privacy with no flow, with stoppage, secrecy, and data minimization” [1].
>
> **Example to differentiate between T1 and T2**
>
> *Response*: Following the explanation in [1], “[Tenet 1] leaves open the question of what it means for flows to be appropriate. To answer, [Tenet 2] introduces the construct of contextual informational norms that express or characterize information flows”.
>
> *Proposed Changes*: We will add a clarification to differentiate T1 and T2 in Section 2.
>
> **Recommendations to overcome hurdles in implementing Tenets**
>
> *Response*: Yes, we will discuss recommendations. CI is a social theory. Hence, we speculate that applying CI to LLMs would require more than a purely algorithmic solution, as you rightly pointed out. Having a “human-in-the-loop” will not be enough, and we require incorporating societal processes such as  governance, policy, and legal institutions, cultural inclusivity, to determine appropriateness of information flows. “Properly” implementing CI would require a multidisciplinary effort (as mentioned in Section 5). This entails explicitly thinking about the role the system plays in the greater context to protect and respect privacy, as well as the sociotechnical implications of their solutions.
>
> We will also highlight recommendations for T2 and T4 which are challenging to implement.
>
> - T2: For gauging  privacy norms, we suggest a deliberative process described in [2] inspired by political science, which involves different expert groups like academic experts, government representatives, and industry professionals in  conversations to reach consensus on social privacy norms. The goal is to assess existing privacy norms, evaluate the impact of new data technologies on these norms, and determine necessary interventions or regulations.
> - T4: Once a breach is flagged in Step 3 (Figure 1), we can use CI Heuristic by deliberation among relevant domain experts on the merits and contribution of the breaching information flow by considering level 1, 2, and 3 analysis as described under T4 in Section 2.
>
> *Proposed Changes*: We will include the above discussion as a new subheading in Section 5. Will this resolve your suggestion?
>
>
> **Typo: CI framework is intuitive and easy-to-implement**
>
> *Response:* This is a typo. We certainly didn’t want to imply that CI is easy to implement. CI framework is intuitive, but not trivial to implement.
>
> *Proposed Changes*: We will update the text to correct this error.
>
> **[Citations]**:
>
> [1] “Nissenbaum, Helen. "Contextual integrity up and down the data food chain." Theoretical inquiries in law 20.1 (2019): 221-256.”
>
> [2] Susser, Daniel, and Matteo Bonotti. "Privacy Mini-Publics: A Deliberative Democratic Approach to Understanding Informational Norms." Annual Symposium on Applications of Contextual Integrity. 2024. Link: https://privaci.info/symposium5/papers/Privacy-Mini-Publics-CI%20Symposium-Daniel%20Susser.pdf)

---

> > ### Comment · Reviewer_NbHH · 2025-04-04
> >
> > Thanks to the authors and I accept the suggestions made by them.

---

### Decision · Program_Chairs · 2025-04-30

**Decision:**

Accept (poster)

**Comment:**

This paper advocates for a more principled and careful use of ideas from Contextual integrity when applying this framework. It demonstrates several prior works that mention using CI in the design but fall short on some of the important aspects of CI. The authors argue that this misuse of CI could give a false impression of a more principled privacy design than there actually is. All in all this work promotes principled and careful understanding of the privacy aspect of an ML system thus is a valuable position to contribute